

# Rock and snow differentiation from colour (RGB) images

Alex Burton-Johnson[1], and Nina Sofia Wyniawskyj[2]

[1]British Antarctic Survey, High Cross, Madingley Road, Cambridge, CB3 OET, UK
[2]Deimos Space UK Ltd., Building R103, Fermi Avenue, Harwell, OX11 0QR, UK.

*Correspondence to:* Alex Burton-Johnson (alerto@bas.ac.uk )

**Abstract**

We present a new method for differentiating snow and rock in colour imagery for application (including by remote sensing non-specialists) to multidisciplinary geospatial analyses in the Polar Regions (e.g. glaciology, geology, and biology). Existing methods for differentiating rock from snow and ice for land cover analysis in the Polar Regions rely on infrared or near-infrared

imagery (e.g. the Normalised Difference Snow Index, NDSI). However, colour images are more abundant and higher resolution. To enable application of this resource, we present and review supervised and unsupervised methods for differentiating rock and snow from colour images. Whilst the unsupervised methods (fuzzy membership and a normalised difference index) are unable to accurately differentiate snow and rock from colour images, supervised classification (Maximum Likelihood Classification (MLC) and a new approach, Polynomial Thresholding (PT)) do achieve high classification

accuracies (95 ±2% for PT and 94 ±3% for MLC, compared with manual delineation). The greater user control of PT achieves better accuracies than MLC in shaded areas (a challenge in high latitudes) and less extensive outcrops. We present the workflow for the new PT method, and provide a calibration tool for its implementation. This approach improves the possible resolution of Polar land cover analysis, and the increases the volume of data that can be utilised.

## 1. Introduction

Delineation of ice and rock extent in glaciated regions is important for navigation, scientific research (e.g. geological and glaciological mapping), and monitoring the environmental response to climate change. However, existing methods rely on infrared (Albert, 2002; Burton-Johnson et al., 2016; Dozier, 1989; Gjermundsen et al., 2011; Hall et al., 1995; Racoviteanu et al., 2010; Raup et al., 2015; Sidjak, 1999; Sirguey et al., 2009), near-infrared, or SAR imagery (synthetic-aperture radar; Atwood et al., 2010), and/or a DEM (digital elevation model; Paul et al., 2002, 2004; Racoviteanu et al., 2010). Colour images

are more abundant and higher resolution, but no method has been evaluated for the differentiation of rock and ice from this data. Suitable existing datasets include aerial and UAV photographs, field photos, and RGB satellite images (e.g. Digital Globe's WorldView, Quickbird, and IKONOS imagery).

The limitations of existing techniques and datasets are illustrated in Antarctica. Prior to Burton-Johnson et al. (2016), the only continental map of rock outcrop was the manually-derived dataset from the Scientific Committee on Antarctic Research

(SCAR) Antarctic Digital Database (ADD) website, www.add.scar.org. This map (Thomson and Cooper, 1993) was derived through manual identification and digitisation of published topographic maps, and suffered from poor georeferencing, frequent misclassification of shaded snow as rock, as well as overestimating and generalising areas of exposed rock. Burton-Johnson et al. (2016) resolved these issues by developing an automated method for rock outcrop classification from multispectral imagery, and published a new continental rock outcrop map for the ADD. Unfortunately, the Landsat imagery used did not

extend south of 82°40' S, and was limited to 30m pixel resolution. A method differentiating rock and snow from colour imagery (particularly airborne and satellite-derived) would enable this resolution and coverage to be improved. Higher resolution would be particularly advantageous for mapping subtle changes in ice extent in response to environmental change, and provide higher resolution basemaps for navigation, geological mapping, and vegetation distribution studies.





We wish to state from the outset that this paper does not intend to address multispectral infrared or near infrared imagery, for which a plethora of methods already exist (e.g. the Normalised Difference Snow Index, NDSI, Dozier 1989). For discussions of these methods, the reader is recommended to refer to Raup *et al.* (2015), Racoviteanu *et al.* (2010), and Albert (2002). Nor do we address vegetated outcrops (our target being high latitudes), or debris-covered glaciers. We accept that it is not feasible to differentiate rock outcrop from debris cover in RGB imagery, but console ourselves that in the Polar Regions (where temperatures are permanently low, and a day-night cycle is absent for much of the year), debris cover is less extensive than at lower latitudes (Fig. 1). Our target is specifically to address the feasibility of rock and snow differentiation in RGB imagery; a subject that has not been addressed, but offers more abundant, higher resolution data than utilised by multispectral techniques. This study includes presentation of a new technique for semi-automated rock outcrop delineation using cloud-free colour (RGB) imagery of glaciated regions: Polynomial Thresholding.

## 2. Methodology

Land cover mapping from remote sensing imagery broadly falls into two classes: 1. Supervised classification - semi-automated methods where the classes are defined and the image analysis algorithm calibrated using manually selected training pixels; and 2. Unsupervised classification - a fully-automated methods where training pixels are not selected prior to image analysis. In the absence of an established method for differentiating rock and snow in RGB imagery, this study evaluates two semi-automated, supervised classification techniques (Maximum Likelihood Classification (MLC), and Polynomial Thresholding (PT)), and two unsupervised classification techniques (Fuzzy Membership (FM), and a normalised difference index (RB-NDSI)).

All of the classification techniques rely on the different relative reflectivities of snow and rock within the visible wavelengths of the electromagnetic spectrum (Fig. 2). For common rock types, red wavelengths are more reflective than blue, whilst snow is more reflective of blue than red wavelengths. However, this difference is significantly smaller than that between the visible wavelengths and the infrared or near-infrared wavelengths (Fig. 2) employed by the NDSI and other multispectral methods for differentiating rock and snow.

To enable broad application, we have selected methods here which can be easily implemented by the reader using the Esri ArcGIS® and ArcMap™ Spatial Analyst toolbox ("Maximum Likelihood Classification" for MLC, "Fuzzy Membership" for FM, and "Raster Calculator" for PT and the RB-NDSI), or similar tools in other GIS software (e.g. QGIS).

### 2.1. Supervised Classification

Both MLC and PT require the user to classify a subset of pixel values in the image as either rock or snow, and then use either image analysis methodology to extrapolate this subset to the whole image. Whilst the requirement for identifying an input "training set" of pixels makes supervised classification slower than unsupervised techniques, it remains a great time saving compared to manual classification (Albert, 2002).

### 2.1.1. Maximum Likelihood Classification (MLC)

MLC is the image classification tool incorporated into Esri ArcGIS® and ArcMap™ (https://desktop.arcgis.com/en/arcmap/latest/tools/spatial-analyst-toolbox/maximum-likelihood-classification.htm). MLC is a popular supervised classification technique, and achieves high classification accuracies (~90%) in differentiating rock and snow from multispectral imagery (Albert, 2002). By assuming that the values of rock and snow pixels are each normally distributed, MLC assigns a mean vector and covariance matrix to each of the two classes. The statistical probability is then





calculated for each pixel, and the most probable class assigned to that pixel. Using the same training pixels, MLC was evaluated using either two or four land cover types (rock and snow, or shaded and sunlit rock and snow).

### 2.1.2. Polynomial Thresholding (PT)

The new PT approach presented here employs the concept that for a given observed intensity of reflectance, rocks will have a
greater red/blue ratio than snow (Fig. 2). With increasing intensity (we use red reflectance here), the red/blue ratio differentiating rock and snow pixels increases along a second order polynomial curve (Fig. 3b).

To exploit this polynomial relationship, a training set of pixel values are manually derived for snow and rock (Fig. 3a), and the second-order polynomial curve separating the two classes derived (Fig. 3b, using the spreadsheet in the Supplementary Material and hosted at https://github.com/Alex-Burton-Johnson/RGB_Rock-Snow_Differentiation). A threshold raster of red
reflectance (a proxy for illumination intensity) is then derived from the red/blue ratio of each pixel using the equation of the polynomial curve. Each pixel is classified as rock or snow (Fig. 3c) according to whether its red reflectance value exceeds this threshold (i.e. classify as snow) or falls below the threshold (i.e. classify as rock).

### 2.2. Unsupervised Classification

Unlike supervised classification techniques, unsupervised classification does not require the user to generate a training set
prior to classification, and is thus less time intensive (Albert, 2002).

### 2.2.1. Fuzzy Membership (FM)

FM transforms the input data on a scale of 0 to 1, based on the probability of each pixel belonging to a determined land cover type (https://desktop.arcgis.com/en/arcmap/10.3/tools/spatial-analyst-toolbox/how-fuzzy-membership-works.htm). There are multiple fuzzy membership transformation algorithms dependent on the assumed distribution of the input data and features of
the land cover types (e.g. whether the land cover type of interests at the upper, lower, or middle of the pixel value ranges). To differentiate snow and rock, we assume that the two land cover types form end members of range, and apply the linear unmixing algorithm to transform (i.e. "unmix") the spectral data into a linear function from 0 to 1. This method assumes that the spectral reflectance of each pixel is a linear combination of the spectra of the two possible end member materials (snow and rock), and each pixel is thus classified along the scale according to its similarities to these end members. Within this 0 to 1 range, the user
then defines a threshold value differentiating snow and rock. The linear unmixing algorithm is the most accurate fuzzy classification for differentiating snow and rock in multispectral data  (Albert, 2002).

### 2.2.2. Normalised Difference Index (RB-NDSI)

The most commonly used method for differentiating snow and rock in multispectral infrared imagery is the Normalised Difference Snow Index (Dozier, 1989; Hall et al., 1995). This approach follows the normalised difference vegetation index
used to map vegetation cover (NDVI; Tucker, 1979, 1986) by normalising the difference between two wavelengths with very different reflectivities from snow and rock (the green and SWIR (Short Wave Infrared) wavelengths; Fig. 2 and Equation 1), and defining an optimal threshold to differentiate the two land cover types.

$$NDSI = \frac{Green - SWIR}{Green + SWIR}$$      (1)

Whilst high accuracies can be achieved at low latitudes (Albert, 2002), the NDSI is unable to differentiate snow and rock in the prevalent and unavoidable shadows of high latitudes, and also misclassifies cloud cover as rock outcrop (Burton-Johnson



et al., 2016). The optimal threshold value may also change between different images, or even across the same image, in response to changes in illumination or fresh snow cover (Burns and Nolin, 2014). Consequently, broad automated application of the NDSI across the Polar Regions (as in the generation of the Landsat 8 Antarctic rock outcrop map; Burton-Johnson et al., 2016) requires combining the NDSI with other thresholds.

Whilst the NDSI uses infrared imagery, we evaluate whether the differential reflectivities of the red and blue wavelengths (Fig. 2) are sufficient for a normalised difference approach using the following equation for a red and blue version of the Normalised Difference Snow Index (RB-NDSI):

$$\text{RB-NDSI} = \frac{Blue - Red}{Blue + Red} \qquad (2)$$

## 3. Accuracy Assessment

To evaluate the accuracy and limitations of the classification methods, rock outcrops were manually digitised in ten 100 x 100 m aerial photographs of 0.14-0.25 m resolution from along the Antarctic Peninsula (Fig. 4, available in the Supplementary Material), selected to represent a range of geology, geomorphology, latitude, and illumination. For the supervised classifications, training pixels were then selected in each image, and each image individually classified as snow or rock using the different classification techniques. Classification accuracy was evaluated using 100,000 randomly distributed points to compare the manually-derived land cover map with the classification outputs. The number of QC sampling points used for each land cover type is proportional to each type's relative area. Classification accuracy was calculated as total classification accuracy ($CA_{Total}$; Equation 3) and classification accuracy of rock outcrop ($CA_{Rock}$; Equation 4):

$$CA_{Total} = \frac{Correct\ snow\ pixels + Correct\ rock\ pixels}{Omission\ rock\ pixels + Commission\ rock\ pixels} \qquad (3)$$

$$CA_{Rock} = \frac{Correct\ rock\ pixels}{Omission\ rock\ pixels + Commission\ rock\ pixels} \qquad (4)$$

Where "Omission rock pixels" are rock pixels in the image that were not classified as rock by the algorithms, and "Commission rock pixels" are snow pixels that were incorrectly classified as rock.

## 3.1. Supervised Classification

Both PT and MLC used the same training pixels, although for MLC these were classified as either two (snow or rock) or four land cover types (shaded or sunlit rock, and shaded or sunlit snow). Pixels were chosen in each image to represent a range of illumination (Fig. 3a), and were particularly selected across the margins between rock and snow, where mixed pixels make differentiation challenging. Both PT and MLC performed well in differentiating snow and rock in all images. Mean total classification accuracy ($CA_{Total}$; Fig. 5a and Equation 3) was 0.95 ±2% (1 Standard Deviation, SD) for PT and 0.94 ±3% for MLC (with two types of land cover). Mean classification accuracy of rock outcrop ($CA_{Rock}$; Fig. 5b and Equation 4) was 0.89 ±4% for PT and 0.86 ±7% for MLC. Increasing the number of land cover types to four for MLC did not increase $CA_{Total}$ (0.94 ±3%; Fig. 5a), and only slightly increased $CA_{Rock}$ (0.87 ±7%; Fig 5b); although this also slightly increased its SD.

Despite training pixels including areas of shade, PT achieved greater accuracies in deeply shaded pixels where MLC using two land cover types incorrectly classified snow as rock (Fig. 6b); increasing its commission error relative to PT (Fig. 5). MLC attributes lower confidence values to its classification of shaded pixels, but reclassifying at least the 50% least confident pixels as snow is required to reduce the commission error sufficiently (Fig. 6). Because rock omission error increases at a greater rate than the rock commission error decreases with this confidence-based pixel reclassification, this approach greatly reduces the overall accuracy of the MLC classification (Fig. 7). Using four land cover types increases the accuracy of MLC in shaded





150    regions compared with using two land cover types, but still incorrectly classifies more snow as rock than the PT technique (Fig. 6c), and only produces a marginal and variable improvement in overall classification accuracy (Fig. 5).

Neither PT nor MLC show a correlation between the number of training pixels used and the resultant accuracy of the classification (Fig. 8a to 8c). What is important for an accurate classification is not so much the number of training pixels, but the range of pixels they sample. High accuracy classification requires sampling of both shaded and illuminated pixels of snow and rock. Because mixed pixels are the hardest to classify, it is particularly important that the training set includes pixels across the margins between snow and rock.

A strong correlation exists between the accuracy of the MLC classification and the proportion of the image composed of rock pixels (Fig. 8e and 8f). This correlation is strongest for MLC with four land cover classes (Fig. 8f), and is much weaker for the PT approach (Fig. 8d). This relationship between rock extent and accuracy reflects the increased inaccuracy along the margin of rock outcrops compared with the centre of the outcrop. Smaller outcrops have a greater proportion of marginal pixels, so are classified at lower accuracies than more extensive outcrops.

### 3.2. Unsupervised Classification

Despite varying the threshold used to differentiate snow and rock from the RB-NDSI method, the approach was unable to provide a satisfactory classification. With too low a threshold, RB-NDSI fails to identify rock outcrop in shaded regions, and increasing the threshold falsely classifies sunlit snow as rock whilst still excluding shaded rock outcrop (Fig. 9). Consequently, we conclude that a normalised difference approach is not applicable to the differentiation of snow and rock from colour imagery.

The fuzzy membership (FM) approach was highly variable in its classification accuracy and optimal threshold value (Fig. 10). Whilst a maximum $CA_{Total}$ of 96% and $CA_{Rock}$ of 93% were achieved for image 'a', the maximum $CA_{Total}$ of images 'd' and 'i' were 74% and 73% respectively. The maximum mean $CA_{Total}$ was 82 ±14%, using a threshold value of 0.2, and whilst the overall accuracy increased with an increasing threshold, the mean commission error when using a threshold value of 0.2 is 15 ±15% (the commission error increasing with an increasing threshold value, Fig. 10). Such a low overall accuracy and high commission error are unsatisfactory when compared with supervised classification techniques, and thus this technique is not recommended.

### 4. Discussion

Neither unsupervised classification technique achieved sufficient accuracy in differentiating snow and rock for us to recommend their application. Whilst both the supervised classification techniques (PT and MLC) achieved high classification accuracies, MLC is more limited in its classification accuracy of shaded pixels, and its inaccuracy along rock outcrop margins results in a strong correlation between outcrop extent and accuracy (Fig. 8). This relationship is weaker and of a lower gradient with PT. The same training set is required for both approaches, and its creation is the most time consuming part of the classification process (allow 5-10 minutes to create the training set, and a further ~5 minutes to set up, calibrate, and implement the algorithms, although the actual processing takes seconds). Therefore, as both approaches take a similar time to complete, we advocate PT as the most accurate method for differentiating rock and snow in colour images.

Whilst the user can only modify the MLC classification based on the training set used and the confidence threshold assigned, the alternative PT approach can be easily calibrated by the user by changing the threshold curve in the supporting calibration spreadsheet. Consequently, when differentiating mixed pixels of rock and snow, the user can choose to be more conservative or liberal in their classification. The methodology is much faster than manual differentiation, and of comparable accuracy (Paul et al., 2013). A range of image illuminations were evaluated in our accuracy assessment (Fig. 4), and high accuracies were achieved in all ($CA_{Total}$ = 0.95 ±2%). This is comparable to the popular NDSI technique using infrared imagery for well-



illuminated images (93.9%, Albert, 2002), and better than the NDSI method in shaded regions, where rock outcrop is not
detected by the NDSI, or clouded images where clouds are incorrectly classified as rock (Burton-Johnson et al., 2016).
Differentiating snow and rock from colour images requires cloud free imagery (a common issue for remote sensing data), and
data acquired during daylight exposure (an obvious but common limitation for earth observation at high latitudes). Particular
care should be taken when applying the PT technique in areas with debris covered glaciers, where debris could be erroneously

classified. Variable snow cover must also be considered when applying these techniques to change detection.

Resolution is the greatest limitation in land cover classification. As such, whilst infrared imagery offers automated
differentiation of snow and rock (Burton-Johnson et al., 2016; Dozier, 1989), the more abundant high resolution colour imagery
offers higher resolution and accuracy in land cover classification in Polar Regions (e.g. aerial photography and high-resolution
satellite imagery, including DigitalGlobe's WorldView, QuickBird, and IKONOS products).

**5. Applications and Future Developments**

The techniques discussed here have a range of possible applications to geoscience and cartography. Being able to accurately
locate rock outcrops is integral to production of navigational and geological maps. Both PT and MLC of colour images may
also aid monitoring of ice extent changes. By only requiring colour images to analyse the relative extents of rock and ice, an
extensive historical dataset can be analysed, allowing changes in these extents over time to be determined (e.g. using the USGS

and BAS Antarctic aerial photography archive, Fig. 11; freely available from https://earthexplorer.usgs.gov). This provides
another method for evaluating the effect of climate change in the Polar Regions, as well as other glaciated areas. The abundance
and resolution of satellite and airborne colour imagery continues to increase, and these new datasets will allow further
application of land cover analysis at higher resolutions and consequently higher accuracies, allowing evaluation of changing
snow and ice extent in response to climate change. By employing higher resolution imagery, more detailed base maps can be

developed for local studies including glaciological, geological, and vegetation mapping. The increasing availability of UAV
photography can be used to develop 3D imagery to which this land cover classification can be applied, allowing high-resolution
3D evaluation of stratigraphy and geomorphology.

**6. Conclusions**

Whilst differentiating rock and snow in remote sensing data is normally achieved using infrared imagery, we have shown that

colour images can also achieve high classification accuracies. The two unsupervised techniques (fuzzy membership and a
normalised difference index) were unable to accurately differentiate snow and rock from RGB images. Nevertheless, the two
supervised classification techniques evaluated (Maximum Likelihood Classification (MLC), and a new method: Polynomial
Thresholding (PT)) do achieve high classification accuracies. When compared with manual delineation of rock and snow, PT
achieves accuracies of 95 ±2%, and MLC achieves 94 ±3%. This is comparable to the commonly-used Normalised Difference

Snow Index (NDSI), but unlike the NDSI these methods do not require infrared imagery, and are able to detect rock outcrops
in shaded regions. However, MLC is less accurate than PT in shaded regions (resulting in false positives in rock identification)
and in images with less extensive rock outcrops. Application of colour imagery to the differentiation of snow and rock greatly
increases the availability and resolution of available data for mapping and monitoring of glacial extent. A calibration
spreadsheet to aid in PT analysis is provided in the Supplementary Material, and hosted at https://github.com/Alex-Burton-

Johnson/RGB_Rock-Snow_Differentiation.



**Acknowledgements**

This study is part of the British Antarctic Survey research programme, funded by the Natural Environmental Research Council. We wish to thank Peter Fretwell, Adrian Fox, and the members of the British Antarctic Survey Mapping and Geographic Information Centre (MAGIC) for providing data access and their assistance in developing this work.

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



**Figures**

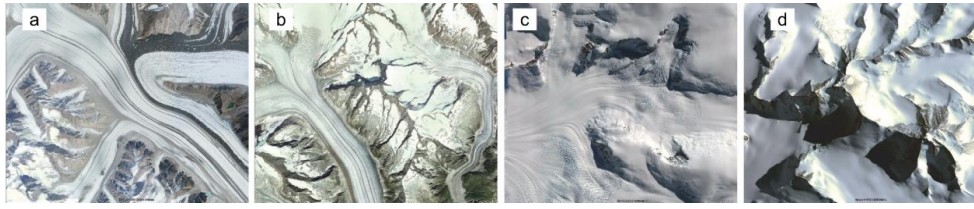

**Figure 1. Comparison of debris cover for glaciers at low latitudes ('a', Karakoram Range (35°N), and 'b', Jungfrau Range, Alps (46°N)) with those of Antarctica ('c', Antarctic Peninsula (66°S), and 'd', Transantarctic Mountains (72°S)). Note the absence of surface moraine and the presence of deep shadows in 'c' and 'd'. This is typical of Antarctic glaciers where a lack of day-night cycle and year-round low temperatures restricts freeze thaw action, and the permanently low sun angles result in deep shadows (from Burton-Johnson et al., 2016). Map data: Google©, Maxar Technologies, NASA, and Landsat / Copernicus.**


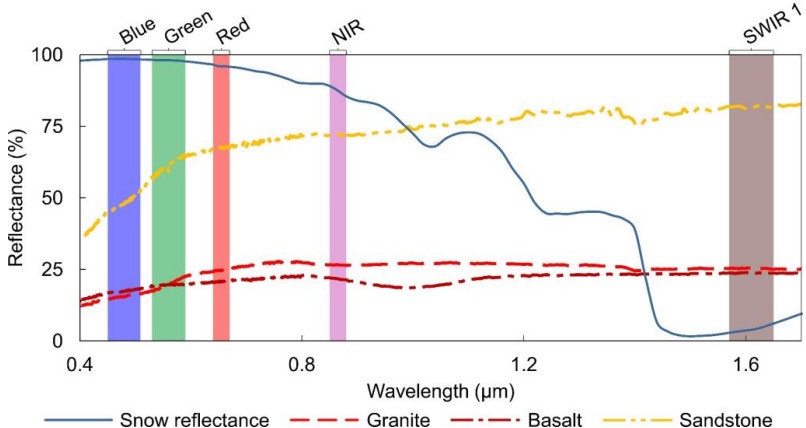

**Figure 2. Spectral reflectance data for snow and rock (granite, basalt and sandstone) from the ASTER Spectral Library v1.2 (Baldridge et al., 2009). Designations of spectral regions as defined by the Landsat 8 bands: Blue – Band 2, 0.45 – 0.51 µm; Green – Band 3, 0.53 – 0.59 µm; Red – Band 4, 0.64 – 0.67 µm; NIR, Near Infrared – Band 5, 0.85 – 0.88 µm; SWIR 1, Short Wave Infrared – Band 6, 1.57 – 1.65 µm.**

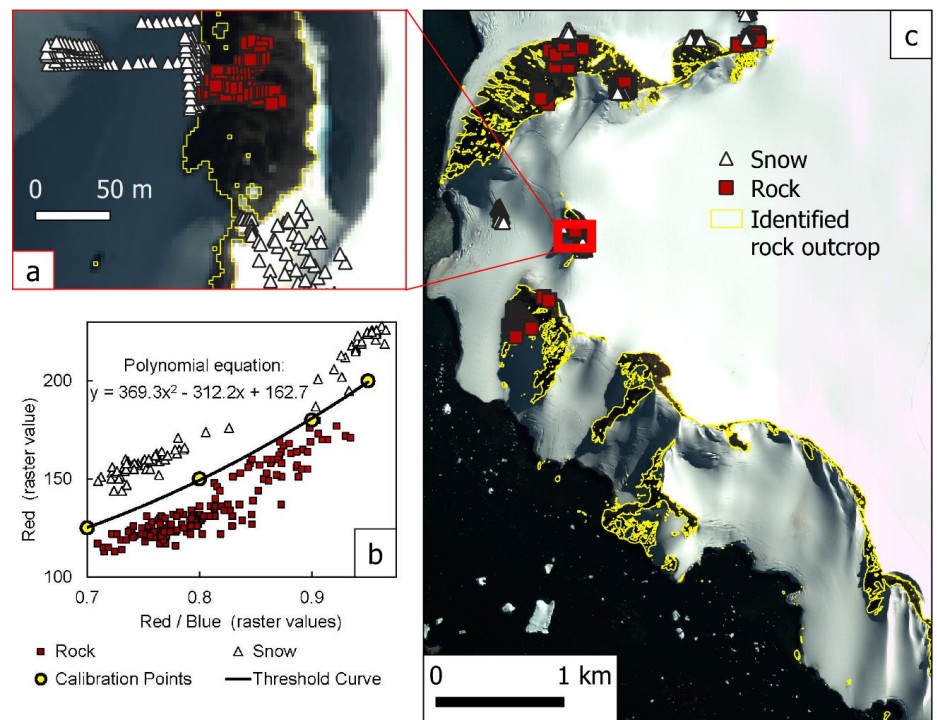

**Figure 3. Illustration of the Polynomial Thresholding workflow, applied to a WorldView RGB image of the Wright Peninsula, Antarctic Peninsula. a) Point shapefile features are created covering pixels of snow and rock in sun and shade, and the raster RGB values extracted to these points. b) The extracted red and blue pixel values are loaded into the calibration spreadsheet (Supplementary Material) and the polynomial curve differentiating the snow and rock pixels calibrated by the user by modifying the "Calibration points" (yellow circles). c) The equation of the polynomial curve is used to calculate a threshold raster of red values from each pixel's red/blue ratio. This threshold is used to identify pixels as snow where there red value exceeds the threshold, or rock when their red value is below the threshold. The identified rock outcrops are outlined in yellow. Imagery courtesy of Digitalglobe Maxar.**





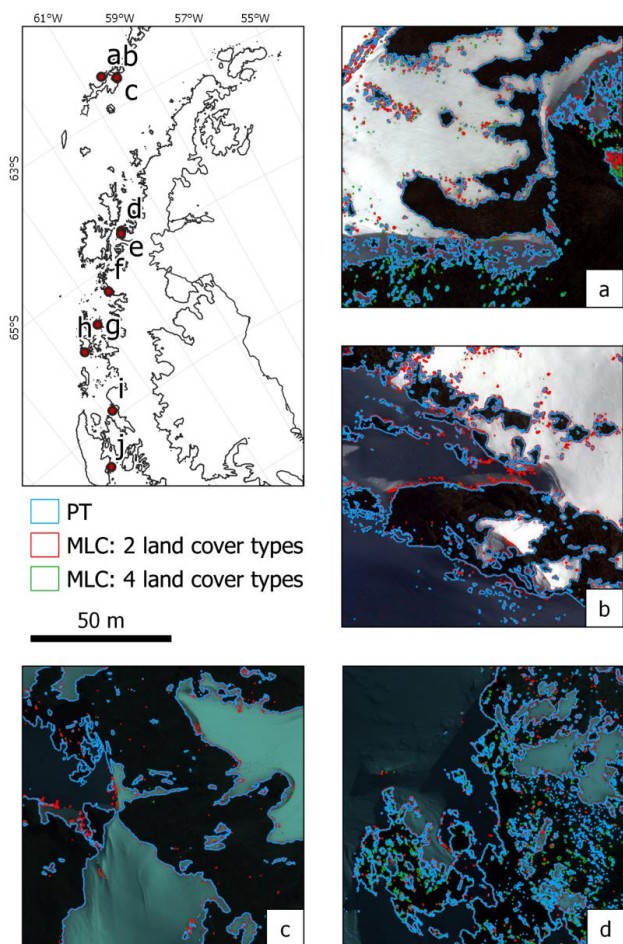



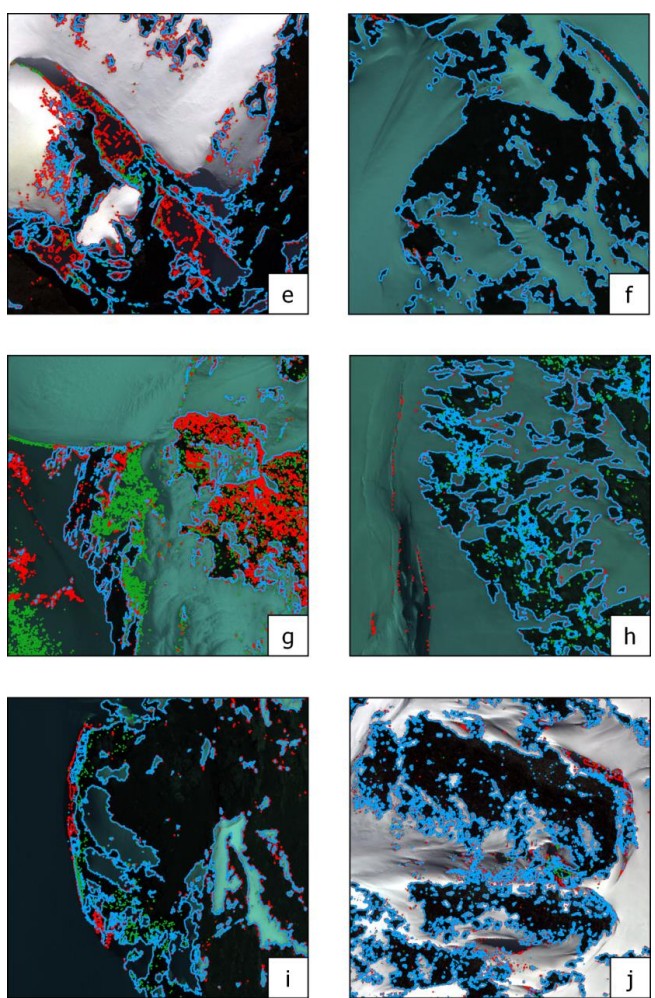

**Figure 4. Ten 100 x 100 m aerial colour photographs of the Antarctic Peninsula (locations shown on inset map) used for accuracy assessment of the PT and MLC methods (MLC using training pixels classified as two or four land cover types; i.e. rock and snow, or sunlit and shaded rock and snow). Rock outcrops identified by PT are outlined in blue, and those identified by MLC are outlined in red (classified using two land cover types) and green (classified using four land cover types). Aerial photography courtesy of the British Antarctic Survey.**





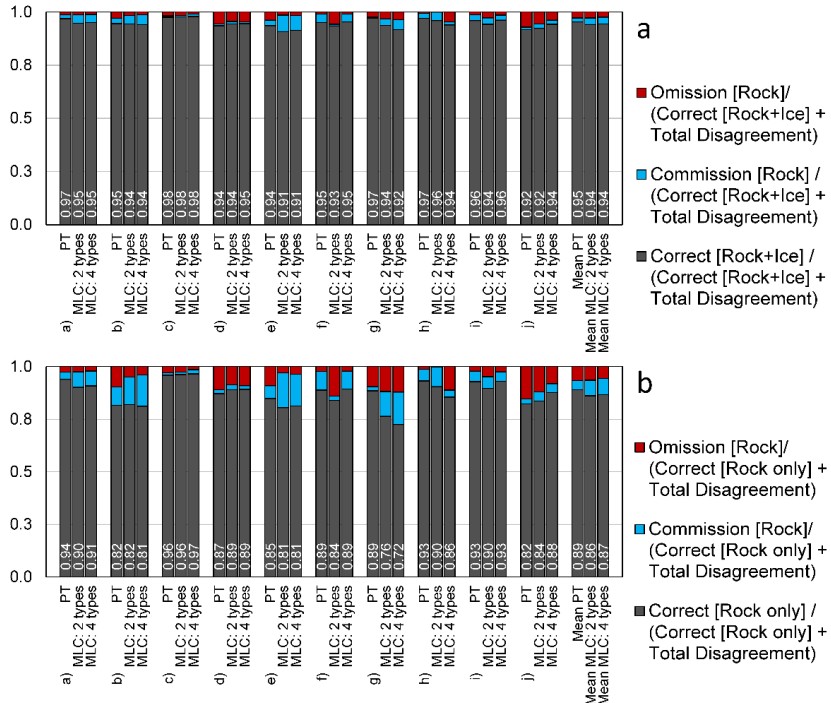

Figure 5. 100% normalised accuracy assessment data for correctly classified pixels and pixels of omission and commission disagreements for the ten images in Fig. 4, as classified by supervised classification (PT and MLC using two or four land cover types). Values in columns are the classification accuracy values: a) Total classification accuracy ($CA_{Total}$; Equation 3), and b) Classification accuracy of rock outcrop ($CA_{Rock}$; Equation 4).

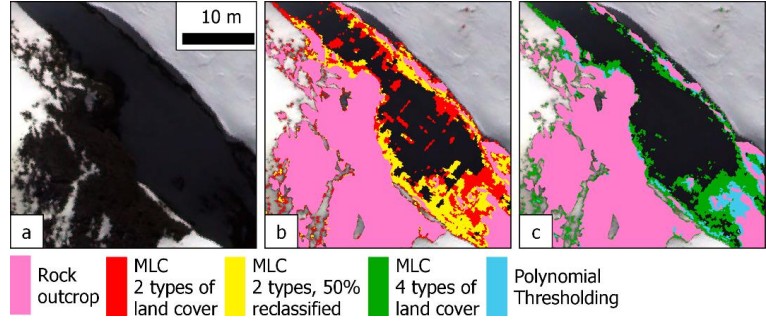

Figure 6. Example of the commission error of MLC, identifying shaded snow as rock (subset of Fig. 4e). a) Original RGB image. b) Manually delineated rock outcrop compared with MLC using two types of land cover (rock and snow). Reclassifying the 50% least confidently classified pixels in the total image as snow ("MLC 50% Reclassified") reduces the commisssion error, but increases the omission error and reduces the total classification accuracy (Fig. 7). c) Manually delineated rock outcrop compared with MLC using four types of land cover (sunlit or shaded snow and rock), and rock outcrop classified by PT. Aerial photography courtesy of the British Antarctic Survey.

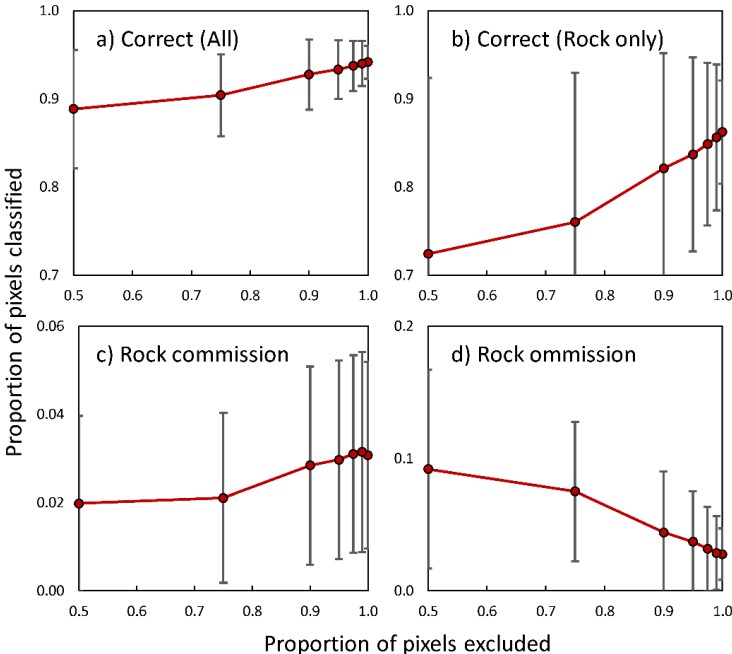

**Figure 7. Effect on MLC accuracy (classified using two types of land cover, snow and rock) of reclassifying a variable proportion of the least confidently classified pixels as snow (based on the confidence values assigned during MLC). a) $CA_{Total}$ (Equation 3). b) $CA_{Rock}$ (Equation 4). c) Commission error (false positives) of rock pixel classification, showing a decrease in error with increased reclassification. d) Omission error (false negatives) of rock pixel classification, showing an increase in error with increased reclassification. The greater increase in omission error (d) with reclassification compared to the decrease in commission error (c) results in the overall decrease in classification accuracy (a and b). The plots are the mean values for all ten images analysed (Fig. 4), with ±1SD error bars on the mean value.**

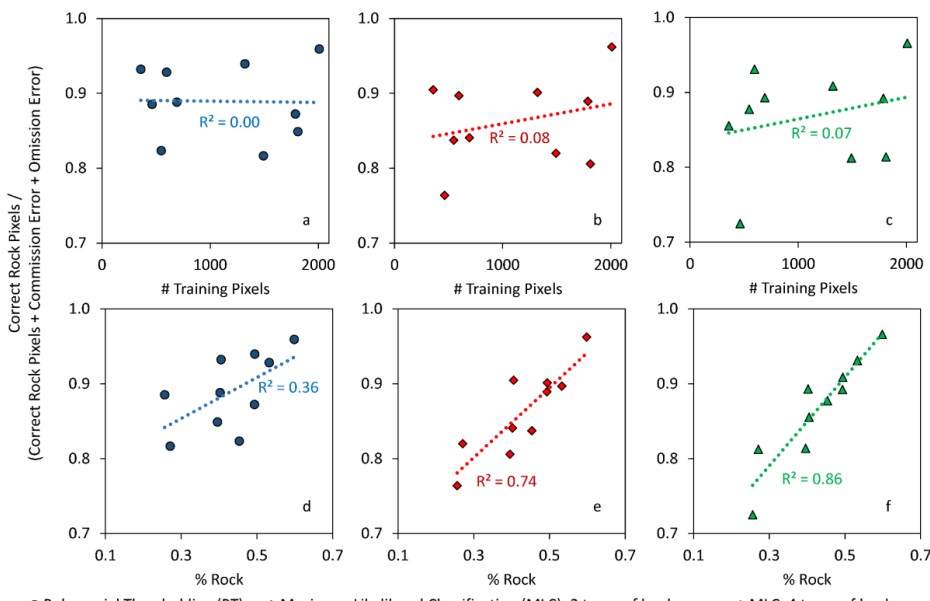

**Figure 8. Comparison for the ten images analysed (Fig. 4) of rock classification accuracy ($CA_{Rock}$; Equation 4) with variation in the number of training pixels, and the proportion of each image occupied by rock outcrop. Linear trendlines and their coefficient of determination ($R^2$) are shown for comparison.**





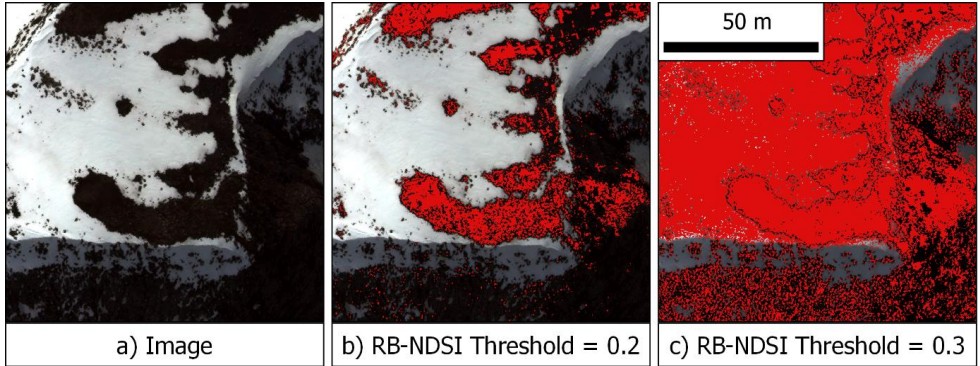

**Figure 9. Application of a red-blue normalised difference snow index (RB-NDSI; Equation 2) to image 'a' from Fig. 4. a) Original image. b) Pixels classified as rock outcrop using an RB-NDSI threshold of 0.2. Note the lack of rock outrop identified in the shaded regions. c) Pixels classified as rock outcrop using an RB-NDSI threshold of 0.3. That increasing the treshold has misclassified the sunlit snow as rock, whilst still ommitting much of the shaded rock outcrop. Aerial photography courtesy of the British Antarctic Survey.**

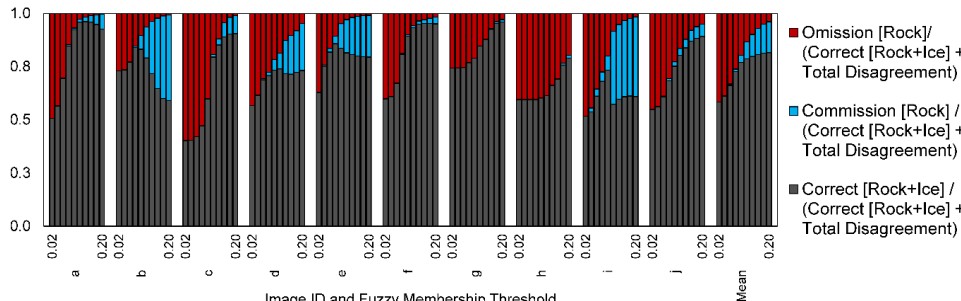

**Figure 10. 100% normalised total classification accuracy assessment data ($CA_{Total}$; Equation 3) for correctly classified pixels and pixels of omission and commission disagreements for the ten images in Fig. 4, as classified by fuzzy membership (FM). Images were analysed using a linear FM algorithm and applied a range of threshold values from 0.02 to 0.2 (x-axis) to differentiate snow and rock.**



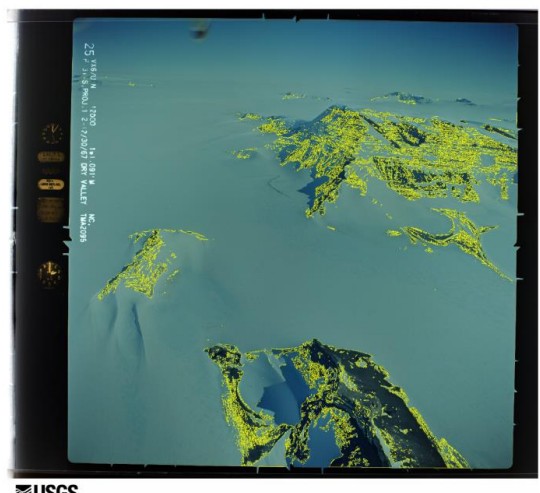

**Figure 11. Application of Polynomial Thresholding to outcrop mapping in archive photography (areas of rock outcrop classifed by PT outlined in yellow). USGS archive image ARCA209531L0025 taken on 30/12/1967 above the Royal Society Range, Antarctica. Note that sky in this oblique image is not a problem for the image classification.**