# Peer review of "Rock and snow differentiation from colour (RGB) images"

_The Cryosphere, 2020_

## Referee Comment (RC1) · Anonymous Referee #1 · 2 Sep 2020

Review of Rock and snow differentiation from colour (RGB) images By Alex Burton-Johnson and Nina Sofia Wyniawskyj

Overall thoughts.

The main contribution is a new very manual PT method of rock/snow classification that can be applied to RGB images. I am not convinced that it really is better than MLC and other standard classifiers out of the ML toolbox. Nevertheless, I think the new PT method is a useful contribution, and a practical tool.

line 80: "the red/blue ratio differentiating rock and snow pixels increases along a second order polynomial curve". This is written as a statement of fact. I think it is an empirical observation that it appears to separate well.

[Figure]

Section 2.1.2: This section should describe how the coefficients of the polynomial determined. I can see from the github excel sheet that in practice it is done by hand tuning. This workflow cannot be fast. I do not think it is disqualifying that you need to manually tune the polynomial, but it has to be very clearly stated here. It is misleading to use the word "derived" for the hand-tuning.

Section 2.1.2: I am convinced that it must be possible to make an algorithm that determines the polynomial coefficients that objectively separates the two classes the best (by some metric). e.g. Using an approach similar to Support Vector Machines (SVM). I think most users would prefer that.

Figure 3: It would make sense to take the logarithm of the intensities before plotting the red vs red/blue. Perhaps the classes could then be separated by a straight line cinstead of a polynomial. The red/blue ratio can easily have outliers in shaded areas. So you can easily end up in a situation where your x-axis is spanning many orders of magnitude. In this plot there are no values with red<100. I.e. there are no pixels that really close to being dark.

Section 2.1.2: I think the PT approach will struggle when blue is near zero.

Section 2.1.1: I suspect that MLC would perform better if the inputs were log-transformed prior to running the algo. Speculation: the distributions will be better approximated by gaussians when you take the logarithm.

Section 3: Question: The performance of supervised methods depend on the relative size of the groups in the training set. Here you write that "training pixels were then selected in each image", but there are little details on how these pixels were selected. Were they selected randomly from the entire image? If so, then that would mean the training set is almost perfectly representative - This would be unlikely to be fulfilled in real applications. Please detail how the training set is selected. And if it is selected by random then please point out what that means. Also: Do you have any thoughts or recommendations for how large the snow vs rock classes should be in the training set.

section 2: There are so many standard supervised classification algorithms from machine learning. You only checked the performance of a single one (MLC) before inventing your own. This is a weakness. I would have liked to see the something like random forests or SVM applied to the problem. At the very least it would be nice to discuss why these ML methods were not considered. These would in principle be less labor intensive than PT as these methods would require zero hand tuning. (Maybe they hav stronger requirements on training data).

line 140: One potential issue with the manual hand tuning is that it can potentially be manipulated to "cheat" in the quality assesment. You can make tiny adjustments to the polynomial coefficients until it performs particularly well on the validation data set. It is concerning that there is this wiggle room, considering that the accuracy difference between PT and MLC are not huge. Let us, for the sake of argument, consider how this could affect the accuracy measures. So hypothetically: We apply the PT approach to a set of images. It generally performs well, but notice that it fails for one image. So we look into why that is. We look at plots like figure 3 and observe that there are two triangles very close to the line near x=0.9. Maybe we should shift the line slightly up so that there is better separation between the main body of points. We make this adjustment and are happy to observe that the performance for this image has improved. This adjustment is entirely sensible, but it will bias the accuracy measure. The validation dataset can no longer be said to have been kept separate from the training.

section 3.2: It is not really surprising that this does not work well. So, in my opinion this does not add much to the manuscript. It would ofcourse be nice to have an algorithm that did not need to be trained as that is labor intensive. However, i think realistically it would be more feasible to use a supervised method where the training is done once and for all or atleast not trained on every single image. E.g. your PT method but with a single global set of coefficients.

Line 157: The performance of supervised methods depend on the relative size of the

C3

**TCD**
[Figure]

groups in the training set. Here you write that "training pixels were then selected in each image", but there are little details on how these pixels were selected. Were they selected randomly from the entire image? If so, then that would mean the training set is almost perfectly representative - This would be unlikely to be fulfilled in real applications. Please detail how the training set is selected. And if it is selected by random then please point out what that means. Also: Do you have any thoughts or recommendations for how large the snow vs rock classes should be in the training set.

I like your PT method even if it is manual. I can see it working with non-calibrated sensors. For traditional NDSI to make fully sense the input has to be corrected for the color of the atmosphere. The PT method has some flexibility. It helps to be able to deal with "fog" between sensor and surface (something like the distance effect in figure 11). It also allows it to deal with images that has been through some auto white balance. Auto white balance, and atmospheric effects would non linearly impact plots like figure 3 and NDSI. I would like to see some discussion of this. Atmospheric effects, white balance, ...

The main difference between RB-NDSI and PT is basically the order of the polynomial as far as i can see. RB-NDSI is just like PT except the line separating classes are a straight line through zero instead of a 2nd degree polynomial. This would suggest that PT would also work for traditional NDSI. Please discuss and speculate this point.

Figures: too many in my opinion.

Figure 4: Can this figure be compressed to a single page. (Suggestion: remove spacing between panels.)

Figure 7: x-axis label. It seems weird that when you exclude 100% then it performs best. Should it be "included" instead of "excluded"?

Figure 7: Here it would be nice if something like this could be done with PT. But because the coefficients are hand tuned, then i dont think that is feasible. (Unfortunately).

Figure 8: Interesting and good point.

figure 9: Not super interesting.

Line 187: I am confused by the citation. This is surely your own result, and so does not require a citation.

Line 183: The hand tuning part of the PT algorithm, and the wiggle room this leaves, means that I am not fully convinced that it is really better than MLC. (I also suspect other ML algos than MLC would perform even better.)

Line 219: I'm pretty sure this difference is not significant.

Line 225: Please include some example datasets in the github excel sheet. For example the points used in figure 3.

---

## Referee Comment (RC2) · Anonymous Referee #2 · 2 Sep 2020

Manuscript Title: "Rock and snow differentiation from colour (RGB) images"

General comments: This study presents a new Polynomial Thresholding (PT) for differentiation of rock and snow using high resolution coloured images of the Polar region. Overall, the work lacks reasonably in scientific content and several fundamental errors. PT is the sole contribution of the current work, however, since the method is empirical it needs rigorous evaluation in terms of its scientific viability. The writing style followed in the manuscript is casual with basic references of certain well-established methods simply missing (For example Linear mixture modelling (LMM) or Fuzzy Membership as called here: Settle and Drake (1993)). The scientific quality of the work needs to be upgraded considerably in line with the comments.

Specific and Technical comments: 1. The Introduction section fails to properly justify

the need of the current work.

2. The results from the proposed PT method are compared with other supervised (MLC) and unsupervised (FM and RB-NDSI) methods. However, this is not a justified comparison except for MLC which performs at-par with the proposed method. Classic NDSI is modified to RB-NDSI as per the availability of the bands and the results of FM are degraded for the sake of comparison. Reframing the NDSI in itself meant that it would not perform well or at par because the spectral difference which is its basis is lacking in its new version (RB-NDSI). Incase of FM or linear mixture modelling neither the method has been properly explained nor correctly implemented. The LMM can be applied in both supervised as well as unsupervised modes and since it is a sub-pixel or soft classifier it results in fraction images equal to the number of target classes (Bastin, 1997). This is usually applied when data has sizable mixed pixels (i.e., in case of moderate to coarse resolution data). Its actual potential is revealed while classifying the low-resolution data and it constitutes an advanced classification method. In this work the output from the LMM has been degraded by thresholding which convert the 'soft' output to a 'hard' one. In light of this the comparisons made with RB-NDSI and FM are not fair and justified.

3. Besides, the authors should have compared their method with the Object based image analysis (OBIA) instead of the methods they have currently chosen. This is because, similar to PT, the OBIA work well in case of high-resolution datasets. It would be interesting to see this comparison.

4. The accuracy assessment portion needs better clarifications and description. Proper explanation and justification in support of the usage of the chosen accuracy metrics (i.e., CAtot and CArock) should be given. Also, for comparison of any set of algorithms the processing speed/time makes a very important parameter which must be considered and which is lacking here.

5. The number of figures can be reduced.

[Figure]

References J. J. SETTLE & N. A. DRAKE (1993) Linear mixing and the estimation of ground cover proportions, International Journal of Remote Sensing, 14:6, 1159-1177, DOI: 10.1080/01431169308904402 L. BASTIN (1997) Comparison of fuzzy c-means classification, linear mixture modelling and MLC probabilities as tools for unmixing coarse pixels, International Journal of Remote Sensing, 18:17, 3629-3648, DOI: 10.1080/014311697216847

---

## Author Comment (AC1) · 13 Nov 2020

Dr Alex Burton-Johnson
British Antarctic Survey
Natural Environment Research Council
High Cross, Madingley Road
Cambridge
CB3 0ET

E-mail: alerto@bas.ac.uk

Dear Anonymous Reviewer RC1,

Thank you for taking the time to provide such a helpful and thorough review. You provided many pertinent comments, and their implementation has greatly improved our manuscript. We have addressed all of your points, and list them below alongside your review.

All the best,

Dr Alex Burton-Johnson

Anonymous Referee #1

Overall thoughts.

The main contribution is a new very manual PT method of rock/snow classification that can be applied to RGB images. I am not convinced that it really is better than MLC and other standard classifiers out of the ML toolbox. Nevertheless, I think the new PT method is a useful contribution, and a practical tool.

- Thank you, we agree. The original manuscript is very open that both PT and MLC are able to achieve similar high accuracies using RGB imagery. Values of 95 ±2% for PT and 94 ±3% for MLC are given in the abstract. The aim of the paper is to show that RGB analysis can be accurate in this application, and quantify its accuracy. The PT method is presented as a new approach, and areas where it should be used in preference to MLC (i.e. shaded regions) discussed.

line 80: "the red/blue ratio differentiating rock and snow pixels increases along a second order polynomial curve". This is written as a statement of fact. I think it is an empirical observation that it appears to separate well.

- Reworded as an empirical observation.

Section 2.1.2: This section should describe how the coefficients of the polynomial determined. I can see from the github excel sheet that in practice it is done by hand tuning. This workflow cannot be fast. I do not think it is disqualifying that you need to manually tune the polynomial, but it has to be very clearly stated here. It is misleading to use the word "derived" for the hand-tuning.

- This process is very swift as all the user needs to do is set the y-axis values of the four points in the supplementary spreadsheet (shown in Fig. 3b) separating the plotted snow and rock values. This is now stated in the manuscript.

Section 2.1.2: I am convinced that it must be possible to make an algorithm that determines the polynomial coefficients that objectively separates the two classes the best (by some metric). e.g. Using an approach similar to Support Vector Machines (SVM). I think most users would prefer that.

- This may be true, but taking away the manual ability to fine tune the threshold will reduce the accuracy and ability for the user to choose more generous or conservative thresholds. This would be a good future project, and one I would be interested in pursuing – thank you for the suggestion.

Figure 3: It would make sense to take the logarithm of the intensities before plotting the red vs red/blue. Perhaps the classes could then be separated by a straight line instead of a polynomial. The red/blue ratio can easily have outliers in shaded areas. So you can easily end up in a situation where your x-axis is spanning many orders of magnitude. In this plot there are no values with red<100. I.e. there are no pixels that really close to being dark.

- Thank you for the suggestion. We have looked into this, but it is unclear how to derive a simple workflow by which the user can calculate the equation required in the way the current method provides. Having reviewed the QC images, a polynomial relationship appears a better fit than a logarithmic one. However, we agree that this is an area that should be explored further.

Section 2.1.2: I think the PT approach will struggle when blue is near zero.

- Exceptionally low blue or red values will only occur in pixels of very low illumination. In these areas, all methods struggle. This study shows that PT achieves higher accuracies than other methods. The NDSI particularly struggles, as shown in my previous publication in The Cryosphere (Burton-Johnson et al, 2016).

Section 2.1.1: I suspect that MLC would perform better if the inputs were log- transformed prior to running the algo. Speculation: the distributions will be better approximated by gaussians when you take the logarithm.

- [As above]: Thank you for the suggestion. We have looked into this, but it is unclear how to derive a simple workflow by which the user can calculate the equation required in the way the current method provides. Having reviewed the QC images, a polynomial relationship appears a better fit than a logarithmic one. However, we agree that this is an area that should be explored further.

Section 3: Question: The performance of supervised methods depend on the relative size of the groups in the training set. Here you write that "training pixels were then selected in each image", but there are little details on how these pixels were selected. Were they selected randomly from the entire image? If so, then that would mean the training set is almost perfectly representative - This would be unlikely to be fulfilled in real applications. Please detail how the training set is selected. And if it is selected by random then please point out what that means. Also: Do you have any thoughts or recommendations for how large the snow vs rock classes should be in the training set.

- The number of training pixels was varied between 350 and 2000 to evaluate this (Fig. 8a-8c in the original manuscript). This is discussed in Section 3.1. of the original manuscript, which concluded that there is no relationship between the number of training pixels and the accuracy of the classification. As noted in the original manuscript, "What is important for an accurate classification is not so much the number of training pixels, but the range of pixels they sample". The number of pixels is now explicitly stated in the text, as well as shown (as previously) in the Figure.

Section 2: There are so many standard supervised classification algorithms from machine learning. You only checked the performance of a single one (MLC) before inventing your own. This is a weakness. I would have liked to see the something like random forests or SVM applied to the problem. At the very least it would be nice to discuss why these ML methods were not considered. These would in principle be less labor intensive than PT as these methods would require zero hand tuning. (Maybe they have stronger requirements on training data).

- In the introduction to Section 2 in the original manuscript, we state that "we have selected methods here which can be easily implemented by the reader using the Esri ArcGIS® and ArcMap™ Spatial Analyst toolbox ("Maximum Likelihood Classification" for MLC, "Fuzzy Membership" for FM, and "Raster Calculator" for PT and the RB-NDSI), or similar tools in other GIS software (e.g. QGIS)." Whilst other methods exist, they require more specialist image analysis software and expertise. The aim of this paper is to enable differentiation of snow and rock by non-specialists in remote sensing, as stated in the opening sentence of the abstract in the original manuscript.

line 140: One potential issue with the manual hand tuning is that it can potentially be manipulated to "cheat" in the quality assesment. You can make tiny adjustments to the polynomial coefficients until

it performs particularly well on the validation data set. It is concerning that there is this wiggle room, considering that the accuracy difference between PT and MLC are not huge. Let us, for the sake of argument, consider how this could affect the accuracy measures. So hypothetically: We apply the PT approach to a set of images. It generally performs well, but notice that it fails for one image. So we look into why that is. We look at plots like figure 3 and observe that there are two triangles very close to the line near x=0.9. Maybe we should shift the line slightly up so that there is better separation between the main body of points. We make this adjustment and are happy to observe that the performance for this image has improved. This adjustment is entirely sensible, but it will bias the accuracy measure. The validation dataset can no longer be said to have been kept separate from the training.

- This ability to manually train the analysis is integral to the PT method, and the cost of this is in the additional time taken over an MLC approach. Consequently, this is not an issue for QC analysis.

section 3.2: It is not really surprising that this does not work well. So, in my opinion this does not add much to the manuscript. It would of course be nice to have an algorithm that did not need to be trained as that is labor intensive. However, i think realistically it would be more feasible to use a supervised method where the training is done once and for all or at least not trained on every single image. E.g. your PT method but with a single global set of coefficients.

- Whilst this does not add much to the manuscript, it is important to address and highlight to the reader – particularly those with limited remote sensing experience.

Line 157: The performance of supervised methods depend on the relative size of the groups in the training set. Here you write that "training pixels were then selected in each image", but there are little details on how these pixels were selected. Were they selected randomly from the entire image? If so, then that would mean the training set is almost perfectly representative - This would be unlikely to be fulfilled in real applications. Please detail how the training set is selected. And if it is selected by random then please point out what that means. Also: Do you have any thoughts or recommendations for how large the snow vs rock classes should be in the training set.

- The training pixels are not selected randomly, but selected to cover pixels representing illuminated and shaded snow and rock across the image. This is now stated explicitly in the introduction to Section 3.

I like your PT method even if it is manual. I can see it working with non-calibrated sensors. For traditional NDSI to make fully sense the input has to be corrected for the color of the atmosphere. The PT method has some flexibility. It helps to be able to deal with "fog" between sensor and surface (something like the distance effect in figure 11). It also allows it to deal with images that has been through some auto white balance. Auto white balance, and atmospheric effects would non linearly impact plots like figure 3 and NDSI. I would like to see some discussion of this. Atmospheric effects, white balance, ...

- Thank you for the support. The comments you suggest have been added to Section 4.

The main difference between RB-NDSI and PT is basically the order of the polynomial as far as i can see. RB-NDSI is just like PT except the line separating classes are a straight line through zero instead of a 2nd degree polynomial. This would suggest that PT would also work for traditional NDSI. Please discuss and speculate this point.

- Thank you for the suggestion, this has now been noted in Section 3.2.

Figures: too many in my opinion.

- As you suggest, we have removed Fig. 9 and have combined Fig. 4a and 4b.

Figure 4: Can this figure be compressed to a single page. (Suggestion: remove spacing between panels.)

- Figure compressed as suggested.

Figure 7: x-axis label. It seems weird that when you exclude 100% then it performs best. Should it be "included" instead of "excluded"?

- Whilst it seems counter-intuitive, we have repeated the analyses and achieve the same result.

Figure 7: Here it would be nice if something like this could be done with PT. But because the coefficients are hand tuned, then i dont think that is feasible. (Unfortunately).

- As you conclude, as similar plot cannot be made for PT due to its workflow.

Figure 8: Interesting and good point.

- Thank you.

Figure 9: Not super interesting.

- Figure removed as suggested.

Line 187: I am confused by the citation. This is surely your own result, and so does not require a citation.

- Paul et al. (2013) evaluated that manual differentiation of snow and rock is ~95%. This is now stated explicitly.

Line 183: The hand tuning part of the PT algorithm, and the wiggle room this leaves, means that I am not fully convinced that it is really better than MLC. (I also suspect other ML algos than MLC would perform even better.)

- As stated implicitly in the original manuscript, we agree based on the results shown here that in well illuminated pixels and away from outcrop margins PT and MLC have similar accuracy. However, when an image includes significant margins and shaded pixels, PT performs better than MLC. As stated in the original manuscript and in our earlier comment, this manuscript only addresses techniques easily employed by non-specialists in commonly used GIS software.

Line 219: I'm pretty sure this difference is not significant.

- Whilst this difference is small, this is because most rock outcrop pixels in the QC analyses are well illuminated and away from rock outcrop margins. The evidence that PT performs better in these pixels is presented in detail in the original manuscript. This is now stated explicitly.

Line 225: Please include some example datasets in the github excel sheet. For example the points used in figure 3.

- Good idea. The calibration spreadsheet used for Image J in Fig. 4 is now included in the Supplementary material and on GitHub. This image was selected as the most useful example for the reader.

---

## Author Comment (AC2) · 13 Nov 2020

Dr Alex Burton-Johnson
British Antarctic Survey
Natural Environment Research Council
High Cross, Madingley Road
Cambridge
CB3 0ET

E-mail: alerto@bas.ac.uk

Dear Anonymous Reviewer RC2,

Thank you for taking the time to provide your review of our submission to *The Cryosphere*. Along with reviewer RC1, implementation of your comments has greatly improved our manuscript. We have addressed all of your points, and list them below alongside your review.

All the best,

Dr Alex Burton-Johnson

Anonymous Referee #2

Manuscript Title: "Rock and snow differentiation from colour (RGB) images"

General comments: This study presents a new Polynomial Thresholding (PT) for differentiation of rock and snow using high resolution coloured images of the Polar region. Overall, the work lacks reasonably in scientific content and several fundamental errors. PT is the sole contribution of the current work, however, since the method is empirical it needs rigorous evaluation in terms of its scientific viability. The writing style followed in the manuscript is casual with basic references of certain well-established methods simply missing (For example Linear mixture modelling (LMM) or Fuzzy Membership as called here: Settle and Drake (1993)). The scientific quality of the work needs to be upgraded considerably in line with the comments.

- Thank you for the reference, this has now been added.

Specific and Technical comments: 1. The Introduction section fails to properly justify the need of the current work.

- On this matter I'm afraid we have to disagree. We state explicitly in the original manuscript the unexploited potential for high resolution classification using RGB imagery; the lack of existing evaluation of this application of RGB data (in contrast to methods using infrared imagery); the application of this data to a range of scientific fields; and the aim of presenting the method to non-specialists in remote sensing. We addressed requests by the subject editor prior to acceptance for peer review to ensure the clarity and aims of the manuscript.

2. The results from the proposed PT method are compared with other supervised (MLC) and unsupervised (FM and RB-NDSI) methods. However, this is not a justified comparison except for MLC which performs at-par with the proposed method. Classic NDSI is modified to RB-NDSI as per the availability of the bands and the results of FM are degraded for the sake of comparison. Reframing the NDSI in itself meant that it would not perform well or at par because the spectral difference which is its basis is lacking in its new version (RB-NDSI). Incase of FM or linear mixture modelling neither the method has been properly explained nor correctly implemented. The LMM can be applied in both supervised as well as unsupervised modes and since it is a subpixel or soft classifier it results in fraction images equal to the number of target classes (Bastin, 1997). This is usually applied when data has sizable mixed pixels (i.e., in case of moderate to coarse resolution data). Its actual potential is revealed while classifying the low-resolution data and it constitutes an advanced classification method. In this work the output from the LMM has been degraded by thresholding which convert the 'soft' output to a 'hard' one. In light of this the comparisons made with RB-NDSI and FM are not fair and justified.

- The target users and aims of the paper are explicitly stated at the start of the original manuscript, requiring a "hard" output of differentiated rock and snow, not a probabilistic "soft" output. Consequently, despite the advanced classification output by the LMM/FM method, the outputs (once converted to the required "hard" classification) are unable to provide the required output required by this study and our stated target audience at sufficient accuracy. We are aware that many other FM classifiers exist (as stated in Section 2.2.1. of the original manuscript), but (as also stated in the original manuscript), Albert (2002) has shown that LMM is the most accurate FM classification method for differentiating snow and rock.

3. Besides, the authors should have compared their method with the Object based image analysis (OBIA) instead of the methods they have currently chosen. This is because, similar to PT, the OBIA work well in case of high-resolution datasets. It would be interesting to see this comparison.

- In the introduction to Section 2 in the original manuscript, we state that "we have selected methods here which can be easily implemented by the reader using the Esri ArcGIS® and ArcMap™ Spatial Analyst toolbox ("Maximum Likelihood Classification" for MLC, "Fuzzy Membership" for FM, and "Raster Calculator" for PT and the RB-NDSI), or similar tools in other GIS software (e.g. QGIS)." Whilst other methods exist, they require more specialist image analysis software and expertise. The aim of this paper is to enable differentiation of snow and rock by non-specialists in remote sensing, as stated in the opening sentence of the abstract in the original manuscript. In light of your comments and RC1, we have expanded on this in the introduction to Section 2 to be even more explicit, with the following text: "We are aware that more sophisticated image analysis and machine learning techniques (e.g. Object Based Image Analysis) can be implemented in specialist remote sensing software packages (e.g. ENVI), but this paper aims specifically to enable non-specialists with a basic background in GIS to quickly and easily derive their required basemap data without further training or software."

4.    The accuracy assessment portion needs better clarifications and description. Proper explanation and justification in support of the usage of the chosen accuracy metrics (i.e., CAtot and CArock) should be given. Also, for comparison of any set of algorithms the processing speed/time makes a very important parameter which must be considered and which is lacking here.

- The justification for using $CA_{Tot}$ and $CA_{Rock}$ have now been added to the introduction of Section 3. The implementation and processing times for MLC and PT are stated in Section 4 of the original manuscript. They of course depend on the size of the image, the number of training pixels, and the processing power of the computer.

5.    The number of figures can be reduced.

- This was also noted by RC1. As suggested by RC1, we have removed Fig. 9 and have combined Fig. 4a and 4b.

References J. J. SETTLE & N. A. DRAKE (1993) Linear mixing and the estimation of ground cover proportions, International Journal of Remote Sensing, 14:6, 1159- 1177, DOI: 10.1080/01431169308904402 L. BASTIN (1997) Comparison of fuzzy c-means classification, linear mixture modelling and MLC probabilities as tools for un-mixing coarse pixels, International Journal of Remote Sensing, 18:17, 3629-3648, DOI: 10.1080/014311697216847